# Color morphing surfaces with effective chemical shielding

Adil Majeed Rather[1], Sravanthi Vallabhuneni [1], Austin J. Pyrch[2], Mohammed Barrubeeah[1], Sreekiran Pillai[1], Arsalan Taassob[1], Felix N. Castellano [2] & Arun Kumar Kota [1] ✉

Color morphing refers to color change in response to an environmental stimulus. Photochromic materials allow color morphing in response to light, but almost all photochromic materials suffer from degradation when exposed to moist/humid environments or harsh chemical environments. One way of overcoming this challenge is by imparting chemical shielding to the color morphing materials via superomniphobicity. However, simultaneously imparting color morphing and superomniphobicity, both surface properties, requires a rational design. In this work, we systematically design color morphing surfaces with superomniphobicity through an appropriate combination of a photochromic dye, a low surface energy material, and a polymer in a suitable solvent (for one-pot synthesis), applied through spray coating (for the desired texture). We also investigate the influence of polymer polarity and material composition on color morphing kinetics and superomniphobicity. Our color morphing surfaces with effective chemical shielding can be designed with a wide variety of photochromic and thermochromic pigments and applied on a wide variety of substrates. We envision that such surfaces will have a wide range of applications including camouflage soldier fabrics/apparel for chem-bio warfare, color morphing soft robots, rewritable color patterns, optical data storage, and ophthalmic sun screening.

Color morphing is the functional mimicry of natural camouflage observed in many species; it refers to color change in response to an environmental stimulus[1–3]. Photochromic materials allow color morphing in response to light[4–7]. While there are a wide variety of photochromic materials, almost all of them suffer from degradation, irreversible photochromism, or poor photochromic fatigue when exposed to moist/humid environments or harsh chemical environments (e.g., acids, bases, oxidizers or organic solvents)[8–11]. One way of overcoming this challenge is by imparting superrepellency (i.e., extreme repellency to liquids)[12–21] to photochromic materials, thereby providing effective chemical shielding. While there are prior reports on imparting superhydrophobicity (i.e., extreme repellency to high surface tension liquids like water)[22–25], to the best of our knowledge, there are no reports on imparting superomniphobicity (i.e., extreme repellency to both high and low surface tension liquids) to photochromic materials. Here, we emphasize that superomniphobic surfaces impart improved chemical resistance compared to superhydrophobic surfaces[16]. However, simultaneously imparting color morphing and superomniphobicity, both surface properties, requires a rational design. In this work, we systematically designed color morphing surfaces with superomniphobicity that display effective chemical shielding against a wide range of liquids (aqueous or organic, acidic or basic, polar or non-polar, etc.). We investigated the influence of polymer polarity and material composition on color morphing kinetics and superomniphobicity. We also demonstrated the universality of the underlying design principles with a wide variety of substrates, as well

[1]Department of Mechanical and Aerospace Engineering, North Carolina State University, Raleigh, NC 27695, USA. [2]Department of Chemistry, North Carolina State University, Raleigh, NC 27695-8204, USA. ✉e-mail: akota2@ncsu.edu

as photochromic and thermochromic dyes. We envision that our color morphing surfaces with effective chemical shielding will have a wide range of applications including camouflage fabrics[26], color morphing soft robots[27,28] rewritable color patterns[29], optical data storage[30,31] and ophthalmic sun screening[32].

## Results and discussion

A careful choice of materials is critical to achieving color morphing and chemical shielding, which are both surface properties. Color morphing (photochromism) in the solid state requires a photochromic dye embedded in a polymer because the polymer acts as an electron donor and improves the coloration efficiency and reversibility[33–35]. Effective chemical shielding via superomniphobicity requires a combination of low surface energy ($\gamma_{sv} < 12$ mN/m) and re-entrant texture (or convex or overhang or undercut or multivalued texture)[16]. Here, it must be noted that a surface is considered super-repellent when the contact angle $\theta^* > 150°$ and sliding angle $\omega < 10°$ (i.e., the minimum tilt angle when a droplet slides off from the surface) for contacting liquids; obtaining low roll-off angles is as important as achieving high contact angles in qualifying the super-repellency of a surface[16,36]. A surface is typically considered superhydrophobic when the contact angle $\theta^* > 150°$ and sliding angle $\omega < 10°$ with high surface tension liquids like water[12,37]; and a surface is typically considered superomniphobic when contact angle $\theta^* > 150°$ and sliding angle $\omega < 10°$ with both high and low surface tension liquids[15,38]. To achieve color morphing and effective chemical shielding simultaneously, we need an appropriate combination of three materials – a photochromic dye, a low surface energy material, and a polymer – in a suitable solvent and a technique that imparts re-entrant texture. In this work, we chose spiropyran (1,3,3-trimethylindolino-6'-nitrobenzopyrylospiran (SP); typically, pale yellow) as the photochromic dye because of its rapid and sustained photochromism[39]. Upon UV irradiation, SP chemically transforms into merocyanine (MC), typically violet or dark purple with absorption at wavelengths between 550 nm and 600 nm via ring-opening cleavage of the C−O bond, resulting in intense absorption bands across the visible region[40] (Fig. 1, Supplementary Note 1 and Supplementary Fig. 1a). We chose fluorocarbon silanes (heptadecafluoro-1,1,2,2-tetrahydrodecyl triethoxysilane (FDTES); $\gamma_{sv} \approx 12$ mN/m) to impart low surface energy[16]. Among fluorocarbon silanes, although chlorosilanes have higher reactivity, we chose an ethoxy silane because hydrolysis of chlorosilanes results in hydrochloric acid, which impedes the SP to MC transformation (Supplementary Note 2 and Supplementary Fig. 2)[41]. We investigated different polymers (polymethyl methacrylate (PMMA), polydimethylsiloxane (PDMS) with varying polarities; here, it must be noted that the polymer serves three functions – binder that holds all the materials together, acts as electron donors and improve the coloration efficiency and reversibility and providing re-entrant texture during fabrication. We chose acetone as the solvent because it dissolves all the above materials for one-pot synthesis. We chose spray coating as the fabrication technique because it is simple, scalable, economical, and provides re-entrant texture (Fig. 1).

To obtain rapid color morphing, choosing a polymer that enhances the color morphing kinetics is essential. Upon UV irradiation of a polymer +SP blend in the solid state, SP chemically transforms into MC. Without the polymer, the kinetics of this chemical transformation in the solid state are sluggish (Supplementary Note 3 and Supplementary Fig. 3). Prior studies have indicated that polymers with polar functional groups display favorable dipole-dipole interactions with the MC form[42,43]. Therefore, we spray coated one-pot solutions of polymer +SP onto glass substrates (see Materials and Methods) using polymers of significantly different Hansen polar solubility parameter[44] $\delta_p$ – polydimethyl siloxane (PDMS, $\delta_p = 2.9$) and polymethyl methacrylate (PMMA, $\delta_p = 10.5$). The Hansen polar solubility parameters of SP ($\delta_p \sim 7.8$) and MC ($\delta_p \sim 7.6$)[45] indicate that they are possibly more compatible with PMMA than PDMS. We investigated the color change of the polymer + SP surfaces in response to 365 nm ultraviolet (UV) irradiation. As anticipated, PMMA + SP blends displayed more rapid increase in absorbance compared to PDMS + SP blends (Fig. 2a). Based on these results, we chose PMMA + SP blends for subsequent experiments because of their rapid color morphing kinetics.

To obtain effective color morphing, it is also essential to understand the influence of PMMA + SP blend composition on color morphing kinetics. To accomplish this, we spray coated different one-pot solutions of PMMA + SP in acetone onto glass substrates (from 20 wt.% to 70 wt.% PMMA; see Materials and Methods). We investigated the color change (from pale yellow to violet) of our PMMA + SP surfaces in response to 365 nm UV irradiation. As the UV irradiation time increased, the absorbance of our PMMA + SP surfaces at 565 nm (measured with UV-Visible spectrophotometry) increased rapidly, eventually reaching a plateau (Fig. 2b, Supplementary Note 1 and Supplementary Fig. 1b). Furthermore, as the PMMA composition increased, the SP to MC transformation rate increased, resulting in a more rapid increase and a higher plateau in absorbance. We modeled the color morphing kinetics of the PMMA + SP surfaces as first-order reaction kinetics, $A(t) = A_p(1 - e^{-kt})$. Here, $A(t)$ is the absorbance at time $t$, $A_p$ is the plateau in absorbance, and $k$ is the rate constant. We chose first-order kinetics because prior reports have indicated that SP to MC transformation is a first-order reaction[46,47]. Our results indicate a reasonable agreement between experimental absorbance measurements and modeling based on first-order kinetics (solid lines in Fig. 2b), with the rate constant $k$ increasing from 0.02/s to 0.04/s, and the plateau in absorbance $A_p$ increasing from 0.55 to 0.8, as the PMMA composition increased from 20 wt.% to 70 wt.%. At PMMA compositions much higher than 70 wt.%, fabrication via spray coating was intractable due to the high viscosity of the solution. Based on these results, we chose 70 wt.% PMMA + 30 wt.% SP blends (Fig. 2c) for subsequent experiments to impart effective chemical shielding. In these 70 wt.% PMMA + 30 wt.% SP blends, we estimated a maximum of 97% conversion of SP to MC after UV exposure (Supplementary Note 1 and Supplementary Fig. 1c). While there was no difference in color morphing (from pale yellow to violet) upon UV exposure at room temperature (-20 °C) or elevated temperature (60 °C), the reverse color morphing

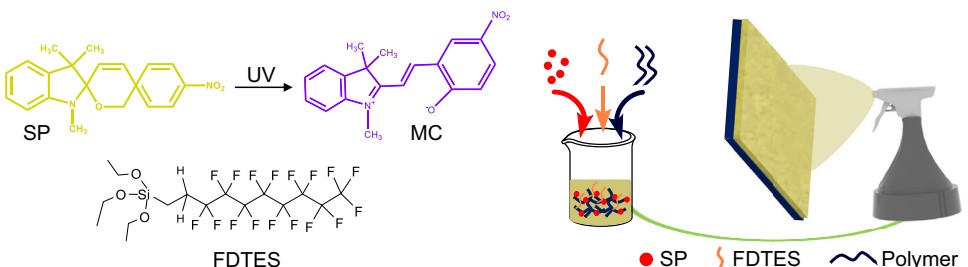

**Fig. 1 | Schematic illustrating the fabrication of color morphing surfaces with effective chemical shielding.** SP is spiropyran. MC is merocyanine. FDTES is heptadecafluoro-1, 1, 2, 2-tetrahydrodecyl triethoxysilane.

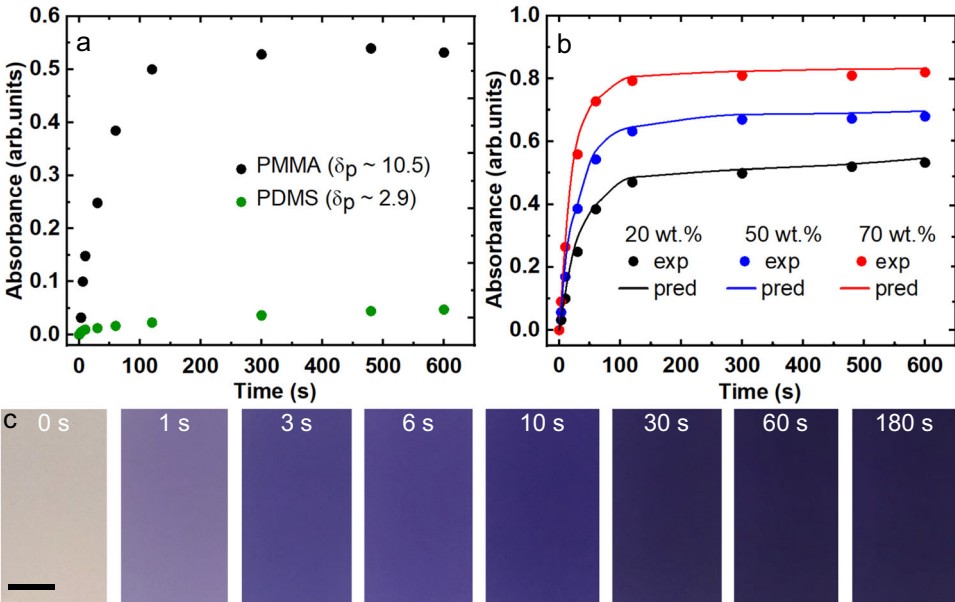

**Fig. 2 | Influence of polymer polarity and composition on color morphing kinetics. a** Absorbance at 565 nm as a function of time for surfaces spray coated with 70 wt.% PMMA + 30 wt.% SP blend and 70 wt.% PDMS + 30 wt.% SP blend. The higher polarity PMMA (Hansen polar solubility parameter, $\delta_p = 10.5$) has faster color morphing kinetics compared to PDMS ($\delta_p = 2.9$). **b** Absorbance at 565 nm as a function of time for surfaces spray coated with PMMA + SP blends with 20 wt.%, 50 wt.%, and 70 wt.% PMMA. Solid dots represent experimental measurements. Solid lines represent predictions based on first-order reaction kinetics. **c** Images of surfaces spray coated with 70 wt.% PMMA + 30 wt.% SP blends after UV exposure at different times. Scale bar represents 5 mm.

(from violet to pale yellow because of MC to SP transformation via ring-closing)[48] required 10 h (half-life, $t_{1/2} \approx 1$ h) at room temperature and 5 min ($t_{1/2} \approx 0.5$ min) at 60 °C (Supplementary Note 4 and Supplementary Fig. 5a, b). Furthermore, the 70 wt.% PMMA + 30 wt.% SP surfaces displayed similar color morphing with 254 nm and 385 nm UV irradiation, as well as with sunlight exposure (Supplementary Note 5 and Supplementary Fig. 6), indicating that our color morphing surfaces are versatile and can be used with a wide range of UV irradiation.

To impart effective chemical shielding to the 70 wt.% PMMA + 30 wt.% SP blends, we added a fluorocarbon silane (FDTES) with low surface energy to the one-pot solution. Effective chemical shielding arises from superomniphobicity, which is a function of both surface texture and surface chemistry (or surface energy)[17]. Here, it must be noted that the addition of FDTES did not appreciably alter the texture of spray coated 70 wt.% PMMA + 30 wt.% SP surfaces (Supplementary Note 6 and Supplementary Fig. 7), but it decreased the surface energy significantly. We evaluated the contact angles and sliding angles of our surfaces using a water ($\gamma_{lv} = 72.1$ mN/m, a representative high surface tension liquid) and hexadecane ($\gamma_{lv} = 27.5$ mN/m, a representative low surface tension liquid). In the absence of FDTES, the 70 wt.% PMMA + 30 wt.% SP surfaces were hydrophobic and oleophilic ($\theta^*_{adv} \approx 120°$, $\theta^*_{rec} \approx 88°$, $\omega \approx 60°$ for 20 μL water droplet, and $\theta^* \approx 0°$, no sliding for 20 μL hexadecane droplet). At lower composition of FDTES (<15 wt.%), the PMMA + SP + FDTES surfaces continued to be hydrophobic and oleophilic ($\theta^*_{adv} \approx 147°$, $\theta^*_{rec} \approx 120°$, $\omega \approx 40°$ for 20 μL water droplet and $\theta^* \approx 0°$, no sliding for 20 μL hexadecane droplet at 15 wt.% FDTES). At intermediate compositions of FDTES (~25 wt.%), the PMMA + SP + FDTES surfaces were superhydrophobic but not superomniphobic ($\theta^*_{adv} \approx 156°$, $\theta^*_{rec} \approx 149°$, $\omega \approx 6°$ for 20 μL water droplet and $\theta^* \approx 118°$, no sliding for 20 μL hexadecane droplet). At higher compositions of FDTES (>35 wt%), the PMMA + SP + FDTES surfaces were superomniphobic ($\theta^*_{adv} \approx 160°$, $\theta^*_{rec} \approx 154°$, $\omega \approx 4°$ for 20 μL water droplet and $\theta^*_{adv} \approx 154°$, $\theta^*_{rec} \approx 146°$, $\omega \approx 7°$ for 20 μL hexadecane droplet at 35 wt.% FDTES) (Fig. 3c, d). Similarly, PMMA + SP + FDTES surfaces were slippery for lower droplet volumes as well ($\omega \approx 5°$ for 5 μL water droplet and $\omega \approx 9°$ for 5 μL hexadecane droplet). This is due to the appropriate combination of re-entrant

texture obtained from PMMA and low solid surface energy obtained from FDTES (Supplementary Note 3 and Supplementary Fig. 4). Scanning electron microscopy (SEM) images confirm the re-entrant texture of PMMA + SP + FDTES blend particles (Fig. 3a). Our theoretical estimations of the contact angles and sliding angles based on the morphology matched reasonably well with the experimental measurements (Supplementary Note 7, Supplementary Note 8 and Supplementary Table 1). Fourier transform infrared spectroscopy (FTIR) of the PMMA + SP + FDTES surfaces (Fig. 3b, Supplementary Note 9 and Supplementary Fig. 8) confirmed the presence of fluorocarbon functional groups (which impart low surface energy) at the surface with strong absorption peaks around 1200 cm⁻¹ corresponding to the –CF, –CF₂ and –CF₃ group stretching, as well as absorption peaks around 575 cm⁻¹, 730 cm⁻¹ and 1120 cm⁻¹ corresponding to the vibration of C–F bond in –CF, –CF₂ and –CF₃ groups[49,50]. Upon UV irradiation, the PMMA + SP + FDTES surfaces simultaneously displayed superomniphobicity and color morphing (Fig. 3e, f and Supplementary Movie 1). There was no significant difference in color morphing kinetics of the PMMA + SP + FDTES surfaces compared to that of the PMMA + SP surfaces (Supplementary Note 10 and Supplementary Fig. 9). Furthermore, this color morphing from pale yellow to violet (in response to UV irradiation) and violet to pale yellow (at room temperature or elevated temperature) was effective for 10 cycles without loss in photochromism or superomniphobicity (Supplementary Note 4 and Supplementary Fig. 5c, d). In this manner, through a rational choice of materials and processing with one-pot synthesis, we fabricated color morphing surfaces with superomniphobicity.

Our color morphing surfaces with effective chemical shielding are suitable for practical applications like camouflage soldier fabrics/apparel with chemical and biological protection. While the color morphing property can evade detection via camouflaging, the liquid repellency can provide chemical shielding against chemical and biological warfare agents. To demonstrate this, we spray coated polyester fabrics with the PMMA + SP + FDTES solutions. SEM images indicated that the fabric surface is uniformly coated with the PMMA + SP + FDTES blends (Fig. 4a, b) without compromising the porosity. Capillary flow porometry indicated that the fabrics retained their breathability even

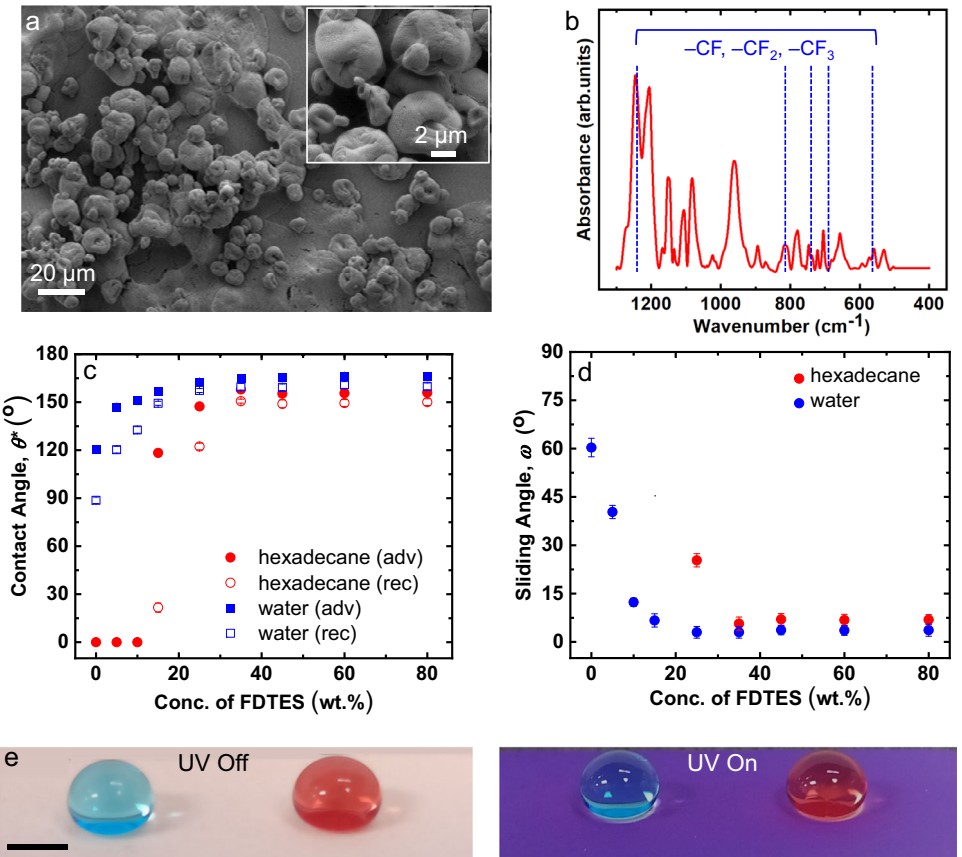

**Fig. 3 | Surface morphology, surface chemistry, liquid repellency, and color morphing. a** SEM image showing re-entrant texture on surfaces spray coated PMMA + SP + FDTES blends. Inset shows an SEM image at a higher magnification. **b** FTIR spectra of PMMA + SP + FDTES surfaces confirming the presence of -CF, -CF$_2$ and -CF$_3$ groups. **c** Advancing and receding contact angles of water and hexadecane on PMMA + SP + FDTES surfaces. Error bars represent SD. **d** Sliding angles of ~20 µl droplets of water and hexadecane on PMMA + SP + FDTES surfaces. Error bars represent SD. **e** Droplets of water (dyed blue with 10 µg/mL methylene blue) and hexadecane (dyed red with 10 µg/mL oil red O) adopting the Cassie-Baxter state and beading up on PMMA + SP + FDTES surfaces before and after exposure to 365 nm UV light. Scale bar represents 5 mm.

after coating (Supplementary Note 11 and Supplementary Fig. 10). As anticipated, the fabric turned from pale yellow to violet upon UV irradiation. To demonstrate the camouflage of the coated fabric, we placed the fabric on a pale-yellow background and gradually moved it to a violet background while simultaneously exposing the fabric to UV light. As the coated fabric transitioned to the violet background, it rapidly turned into the violet color and blended with the background demonstrating effective camouflage (Fig. 4c and Supplementary Movie 2). Both before and after the color morphing, the coated fabric was superomniphobic ($\theta^{*}_{adv} \approx 165°$, $\theta^{*}_{rec} \approx 159°$, $\omega \approx 3°$ for 20 µL water droplet and $\theta^{*}_{adv} \approx 157°$, $\theta^{*}_{rec} \approx 149°$, $\omega \approx 7°$ for 20 µL hexadecane droplet). The chemical shielding due to superomniphobicity is also evident from liquid droplets bouncing on the surface (Fig. 4e, f and Supplementary Movie 3), as well as the resistance to damage (i.e., loss in color morphing or liquid repellency) on exposure to air with 50% relative humidity, immersion in water, exposure to corrosive liquids with a wide range of pH values (acidic, neutral, and basic; Supplementary Note 12, Supplementary Fig. 11 and Supplementary Movie 4) and exposure to oxidizers (30% H$_2$O$_2$; Supplementary Movie 5). Similarly, a wide variety of smooth substrates (e.g., glass, acrylic, PET, aluminum, copper and tin) and rough substrates (paper and fabrics) can be coated with the PMMA + SP + FDTES blends to simultaneously render them with effective camouflage and chemical shielding (Fig. 4g–l, Supplementary Note 13 and Supplementary Table 2).

While our demonstrations thus far relied on the fabrication of color morphing superomniphobic surfaces with a specific photochromic dye, the underlying design principles can be easily extended

to other dyes. To demonstrate the versatility of our technique, we have modified the surface of various commercially available photochromic and thermochromic pigments with FDTES in one-pot synthesis and spray coated them onto polyester fabrics to render them super-omniphobic (Fig. 5a–f, Supplementary Note 14 and Supplementary Table 3) and simultaneously photochromic or thermochromic. The superomniphobicity arises from a combination of the re-entrant texture imparted by the pigment particles and the low surface energy imparted by FDTES. For the photochromic superomniphobic fabrics and the thermochromic superomniphobic fabrics, the dyes can be carefully chosen to tailor the colors (i.e., initial and final colors) and color morphing kinetics upon UV irradiation and heat, respectively (Fig. 5a–f and Supplementary Movies 6 and 7). This color morphing was effective for hundreds of cycles without loss in photochromism or superomniphobicity (Supplementary Note 14 and Supplementary Fig. 12). Furthermore, by spray coating with a shadow mask of the desired pattern on the substrate, photochromic superomniphobic surfaces with cryptic patterns can also be fabricated. The pattern (e.g., "NC STATE") is encrypted (invisible) prior to UV irradiation, decrypted (or revealed) upon UV irradiation, and encrypted again (or erased) upon turning off the UV irradiation (Fig. 5g and Supplementary Movie 8).

In summary, we fabricated color morphing surfaces with effective chemical shielding by spray coating one-pot solutions of PMMA + SP + FDTES blends. We demonstrated that higher polymer polarity leads to faster color morphing kinetics, and when combined with an appropriate combination of re-entrant texture and low surface energy,

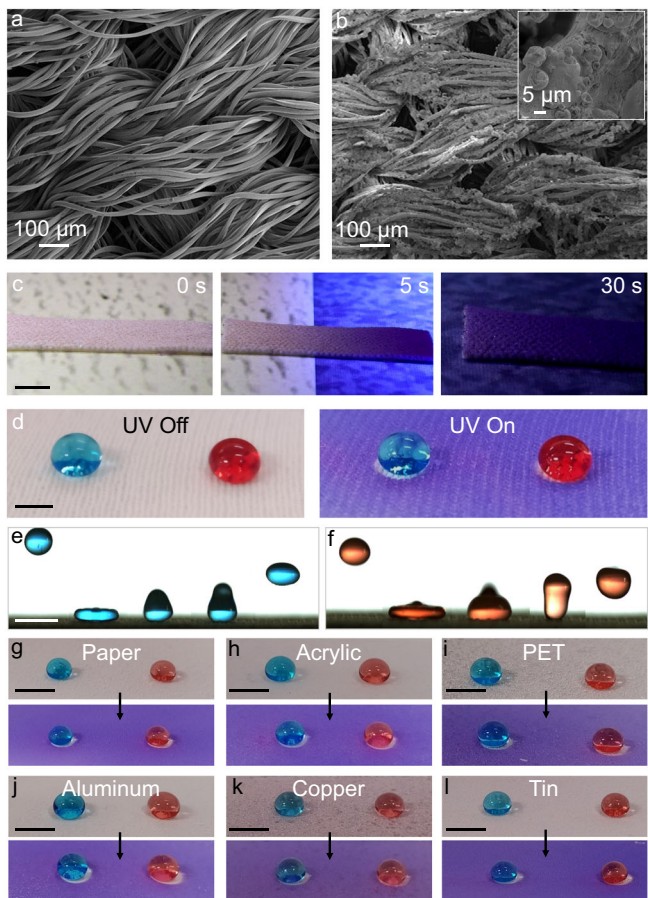

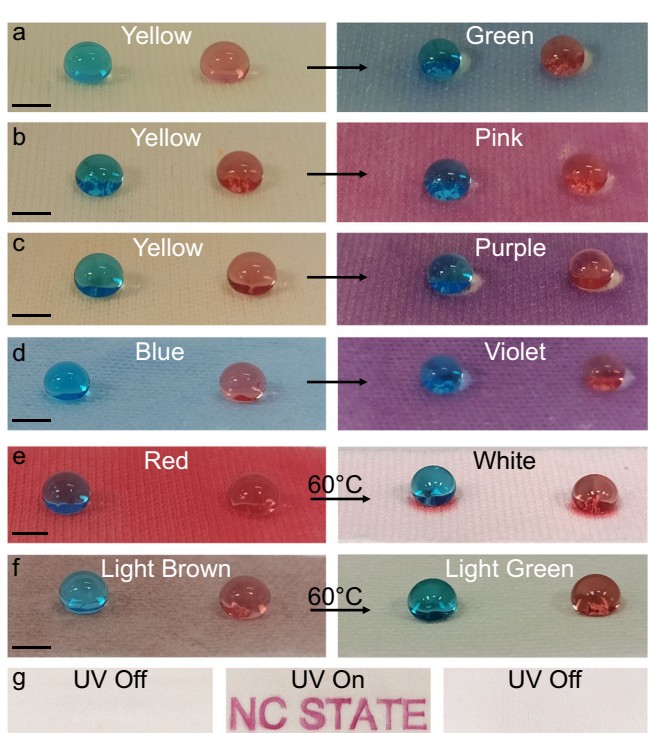

**Fig. 4 | Simultaneous color morphing and liquid repellency on different substrates. a, b** SEM images of uncoated polyester fabric and the fabric spray coated with PMMA + SP + FDTES blend, respectively. Inset in b shows a higher magnification image. **c** Images depicting the effective camouflage of the spray coated fabric upon exposure to 365 nm UV light. **d** Droplets of water (dyed blue with 10 μg/mL methylene blue) and hexadecane (dyed red with 10 μg/mL oil red O) adopting the Cassie-Baxter state and beading up on the spray coated fabric before and after exposure to 365 nm UV light. **e, f** Series of images illustrating droplets of water (blue) and hexadecane (red) bouncing on the spray coated fabric. **g–l** Droplets of water (blue) and hexadecane (red) adopting the Cassie-Baxter state and beading up on different substrates spray coated with PMMA + SP + FDTES blend before and after exposure to 365 nm UV light. Scale bars represent 5 mm for **c–l**.

**Fig. 5 | Photochromic and thermochromic superomniphobic surfaces. a–d** Droplets of water (dyed blue with 10 μg/mL methylene blue) and hexadecane (dyed red with 10 μg/mL oil red O) adopting the Cassie-Baxter state and beading up on superomniphobic surfaces consisting of different photochromic pigments and FDTES before and after exposure to 365 nm light. **e, f** Droplets of water (blue) and hexadecane (red) adopting the Cassie-Baxter state and beading up on superomniphobic surfaces consisting of different thermochromic pigments and FDTES before and after heating. **g** Images illustrating the photochromic superomniphobic fabric with the "NC STATE" pattern encrypted (invisible) before UV irradiation, decrypted (or revealed) upon UV irradiation, and encrypted again (or erased) upon turning off the UV irradiation. Scale bars represent 5 mm for **a–g**.

it can lead to superomniphobicity simultaneously. We demonstrated that the color morphing property of these surfaces enables camouflaging while simultaneously providing effective chemical shielding. Furthermore, we demonstrated that the underlying materials design principles can be extended to fabricate similar surfaces, including cryptic patterns, on a wide variety of substrates and using a wide range of photochromic and thermochromic pigments. While our surfaces can retain both photochromism and superomniphobicity against liquids (Supplementary Note 12 and Note 15, and Supplementary Figs. 11 and 13), further efforts are required to improve their durability against solid abrasion. We envision that our color morphing surfaces with effective chemical shielding will have a wide span of applications, including camouflage solider fabrics, color morphing soft robots, cryptic patterns, optical data storage, and ophthalmic sun screening.

## Methods

### Fabrication of color morphing surfaces
First, glass surfaces and polyester fabric were cleaned with acetone and ethanol and dried with nitrogen gas. Next, one pot solution of (20 wt.% to 70 wt.%) polymethylmethacrylate (PMMA, Mw ~35,000 Da; Scientific polymer) +1,3,3-trimethylindolino-6′-nitrobenzopyrylospiran (SP; TCI America) was prepared in acetone under magnetic stirring. Next, the solution was spray coated (Paasche) on glass slides (Fisher) and polyester fabric (Contec Inc.), and subsequently dried at ambient conditions. Similarly, one pot solution of 20 wt.% PDMS (Smooth-on) +SP was spray coated on glass surfaces and dried at ambient conditions.

### Fabrication of color morphing superomniphobic surfaces
All substrates were cleaned with acetone and ethanol and dried with nitrogen gas. To one pot solution of 70 wt.% PMMA + SP in acetone, 0 to 50 wt.% of heptadecafluoro−1, 1, 2, 2-tetrahydrodecyl triethoxysilane (FDTES; Gelest) was added. Then, the solution was spray coated on glass and polyester fabric surfaces and dried at ambient conditions. Similarly, one pot solution of PMMA + SP + FDTES was spray coated on different substrates, including acrylic (McMaster), PET (McMaster), aluminum (Fisher), copper (McMaster), tin (McMaster), and filter paper (Fisher) and dried at ambient conditions. Furthermore, one pot solution of 50 wt.% commercially available photochromic pigments (B07123TT3X, Uniglow Pigments) + FDTES and 50 wt.% commercially available thermochromic pigments (B07XB51DL4 and B07D1RNHJF, Atlanta Chemical Engineering) + FDTES were prepared in hexane. Then, the solution was spray coated on polyester fabric and dried at ambient conditions.

### Fabrication of patterned color morphing superomniphobic surfaces
First, the polyester fabric was rinsed with ethanol and acetone and dried under nitrogen gas. Next, one pot solution of 50 wt.%

photochromic pigment + FDTES in hexane (Fisher) was spray coated through a shadow mask of the desired pattern (e.g., "NC STATE") on the polyester fabric. Subsequently, the patterned fabric was dried at ambient conditions.

## Color morphing with UV irradiation and heat

Color morphing of photochromic surfaces was performed by irradiating with a 365 nm UV torch (Tattu), as well as 265 nm and 385 nm UV lights (UVP). Color morphing of thermochromic surfaces was performed by heating to 60 °C using a hot plate (Fisher) and cooling back to ambient temperature.

## Contact angle and sliding angle measurements

Contact angles and sliding angles of different liquids were measured using a goniometer (Ramé-Hart 260-F4). Contact angles were measured by advancing or receding a small volume of liquid (~20 μL) onto the surface using a micrometer syringe (Gilmont). Sliding angles were measured by slowly tilting the stage until the droplet (~20 μL or ~5 μL) rolled off from the surface. All results are the average of three individual measurements.

## Morphology characterization

Surface morphology was characterized using a scanning electron microscope (SEM; Thermofisher Phenom Pharos). Images of the color morphing superomniphobic surfaces were obtained using Phenom Pharos Desktop SEM at 15 kV. The samples were sputter coated with a thin film of gold prior to imaging (Cressington 108).

## FTIR characterization

Samples for FTIR (Thermo Scientific Nicolet iS50) were prepared by mixing the desired material with KBr powder to form a pellet. Samples were analyzed with DTGS/KBr detector, and spectra were recorded at $2\,cm^{-1}$ resolution with 32 scans. Background spectra were obtained with an empty pellet holder.

## Bouncing of droplets

Movies of colored water (methylene blue, 10 μg/mL) and hexadecane (oil red O, 10 μg/mL) bouncing droplets were obtained using a high-speed camera (Fastcam Mini AX200) at 500 frames per second.

## Absorption spectra with UV-Visible spectrophotometer

Absorption spectra (400 nm to 800 nm) of the color morphing surfaces were measured with a spectrophotometer (Shimadzu UV-3600). SP to MC conversion was analyzed after irradiation with 365 nm UV. Bare glass slides were used for baseline correction.

## Data availability

The data supporting the key findings of this study are available within the article and the Supplementary Information. Additional data are available from the corresponding author upon request.

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

## Acknowledgements
A.K.K gratefully acknowledges financial support under award 1947454 from the National Science Foundation and under award R21EB033960 from the National Institutes of Health.

## Author contributions
A.K.K. conceived the idea. A.M.R., S.V., A.J.P., M.B., S.P. and A.T. conducted the experiments. A.M.R., S.V., A.J.P., M.B., S.P., A.T. and A.K.K. conducted the analysis. A.M.R., S.V., A.J.P., M.B., S.P., A.T., F.N.C. and A.K.K. wrote the manuscript.

## Competing interests
The authors declare no competing interests.
