## [Peer Review File · Nature Communications]

REVIEWER COMMENTS

Reviewer #1 (Remarks to the Author):

[Note from the Editor: Please see attached PDF]

Please refer to my report in the Review Attachments.

Reviewer #2 (Remarks to the Author):

This manuscript reported color morphing surfaces with effective chemical shielding. The authors carried out the studies about the effects of polymer polarity, PMMA content and FDTES content, etc. After careful reading of this manuscript, I do not think it can be accepted for publication in the journal mainly because of its low novelty as listed below. Instead, I think Scientific Report is a suitable journal for the study.

1. The authors mentioned in the Introduction that “Prior studies have indicated that polymers with polar functional groups display favorable dipole-dipole interactions with the MC form.^{39,40}”. So, it is easy to combine polar polymers like PMMA with SP. The authors did not show anything new regarding the photochromic materials.
2. The authors mentioned in the Introduction that “While there are prior reports on imparting superhydrophobicity (i.e., extreme repellency to high surface tension liquids like water),²²⁻²⁵ to the best of our knowledge, there are no reports on imparting superomniphobicity (i.e., extreme repellency to both high and low surface tension liquids) to photochromic materials.” After careful reading of the references, I found that Ref. 23 has already reported superomniphobic photochromic materials. The superamphiphobicity was tested using much smaller droplets of 5 μL compared with this work.
3. I found that the authors carried most of their studies on rough substrates like fabrics and paper. This is unreasonable to carry out such studies on rough substrates. It is well known in the field that it is easy to form superamphiphobic coatings on rough substrates compared with flat ones like glass slides.
4. During measurements of superamphiphobicity, the authors used very large droplet volume of 20 μL . It is well known again in the field that larger droplet volume is helpful to record lower sliding angle. Such large volume is seldom used in the field.
5. It is also well known that spray-coating can easily form superamphiphobic coatings on various substrates (e.g., Applied Surface Science 2017, 400, 162).

6. Moreover, there is no in-depth discussion and new finding regarding superamphiphobic materials and photochromic materials throughout the manuscript.

There are also other concerns as shown below, which may be helpful for the authors to improve the manuscript before submitting it elsewhere.

1. The authors mentioned in the abstract that “but almost all photochromic materials suffer from degradation when exposed to moist/humid environments or harsh chemical environments”. However, the authors did not perform any study about the resistance of their photochromic materials when exposed to moist/humid environments.
2. It is clear that the fabrics were coated with a lot of the coating materials (Figure 4b). This will inevitably affect the breathability.
3. How about stability of the photochromic materials? Is there any changes in the color parameters and photochromic dynamics after for example hundreds of coloring-decoloring cycles?
4. How about stability of the superamphiphobicity? As a superamphiphobic coating on fabric, Martindale abrasion and washing stability tests are essential.

Reviewer #3 (Remarks to the Author):

This is a well written manuscript on a really interesting topic. Photochromic and thermochromic dyes have been utilized for a wide variety of applications for many years. More recently, the field has been particularly interested in the design and utilization of polar dyes, such as spiropyran used here, as such dyes can sustain their photochromic properties, and may also display a faster response time. The challenge with such dyes is that they can readily lose their performance when they are utilized in an environment where other ions / polar molecules are present. This is the exact challenge that this work aims to address via the utilization of omniphobic surfaces, and this is a major achievement in this field in my opinion.

The work performed is easy to understand, and is well supported by fundamental performance / data analysis. I have a few minor issues that should be addressed:

1. It would be good to report the Hansen polar solubility parameter for both SP and MC in order to compare with PDMS and PMMA.

2. The commercial specific photochromic / thermochromic dyes utilized for the experiments shown in Fig. 5 were not clear to me. Those should be clarified in the text and added to the materials and methods section.

This manuscript by Kota and coworkers described a system which is able to achieve effective color morphing while displaying chemical shielding against various environmental stresses. I would recommend this work to be published in Nature Communications if the following concerns can be properly addressed.

1. Photoswitching of SP to MC in solution state (ideally in acetone) in the presence and absence of polymers should be described in order to compare with the solid-state behavior.
2. If possible, the switched percentage (MC% in PSS) should be measured. Also, their thermal half-lives at ambient or elevated temperature should be provided (e.g. Section S5).
3. What is the loading of SP? Only “70 wt. % PMMA + SP” was mentioned, and the exact composition is unclear.
4. It was shown that the thermal-back reaction could be performed at 60°C, but is it possible to carry out the UV-induced switching also at high temperature (e.g. Section S5)? This experiment would give a more complete picture of the thermal robustness.
5. Would the chemical shielding resist oxidative conditions (such as a H₂O₂ solution)?
6. Regarding the fatigue resistance of this system, the authors described that the color morphing remained effective for 10 cycles, but no experimental data was provided in the SI.
7. In Figure 5 for example, the water and hexadecane have obviously been dyed. The identity and concentration of these dyes should be given in the Methods section.
8. What are the identities of thermochromic pigments utilized for the studies in Figure 5?
9. In Figure 1, the chemical structure of FDTES should be drawn in the correct orientation and with better resolution.

Response to Reviewer#1

This manuscript by Kota and coworkers described a system which is able to achieve effective color morphing while displaying chemical shielding against various environmental stresses. I would recommend this work to be published in Nature Communications if the following concerns can be properly addressed.

We thank the reviewer for carefully reading our manuscript, recognizing the novelty and potentially supporting the publication of our work in *Nature Communications*. We have now substantially revised the manuscript based on the reviewer's suggestions. We sincerely hope that the reviewer will find our revisions satisfactory.

1. Photoswitching of SP to MC in solution state (ideally in acetone) in the presence and absence of polymers should be described in order to compare with the solid-state behavior.

We thank the reviewer for this insightful comment on comparing photochromic behavior in the solid state and liquid state. We have now conducted a series of additional experiments to evaluate the color change of 1,3,3-trimethylindolino-6'-nitrobenzopyrylospiran (SP), with and without polymethyl methacrylate (PMMA), in both the liquid state and the solid state.

SP (solid state): The as-received SP in the solid-state appears pale yellow. Upon exposure to UV, there is no change in color. This indicates that SP by itself does not display photochromic behavior in the solid state.

SP in acetone (liquid state): Upon adding SP to acetone, the color of the solution turns violet immediately. Upon exposure of this SP in acetone solution to UV, there is no further change in color.

SP in acetone, spray coated (solid state):

Upon spray coating the solution of SP in acetone on a substrate, the spray coated substrate appears pale yellow in the solid state. Upon exposure of this spray coated substrate to UV, the color change is neither rapid nor intense. This indicates that SP in acetone, after spray coating, displays poor color morphing with low color intensity and slow kinetics.

SP + PMMA in acetone (liquid state):

Upon adding SP and PMMA to acetone, the color of the solution turns violet immediately. Upon exposure of this SP + PMMA in acetone solution to UV, there is no further change in color.

SP + PMMA in acetone, spray coated (solid state):

Upon spray coating the solution of SP + PMMA in acetone on a substrate, the spray coated substrate appears pale yellow in the solid state. Upon exposure of this spray coated substrate to UV, there is a rapid and significant color change to violet, indicating the chemical transformation of SP into Merocyanine (MC). This indicates that SP + PMMA in acetone, after spray coating, displays significant photochromic behavior in the solid state (see Figure 2c).

We have now modified the manuscript and supporting information Section S3 to include these results.

2. If possible, the switched percentage (MC% in PSS) should be measured.

We thank the reviewer for this comment about MC% in PMMA after UV exposure. For our 70 wt.% PMMA + 30 wt.% SP blends, based on the absorbance at 565 nm before and after UV exposure, we have now estimated a maximum of 97% conversion of SP to MC.

We have now modified the manuscript and supporting information Section S1 to include these results.

Also, their thermal half-lives at ambient or elevated temperature should be provided (e.g. Section S5).

We thank the reviewer for this comment. For MC to SP conversion, which is a first order reaction (Richert et al., *Macromolecules*, 1997; Ren et al., *Sensors*, 2007; Bao et al., *J Polym Res.*, 2014; Zhang et al., *J. Mater. Chem. C*, 2020; Wang et al., *J. Am. Chem. Soc.*, 2022), we have now estimated the half-lives at ambient and elevated temperatures to be ~1 hour and ~0.5 minute, respectively.

We have now modified the manuscript and supporting information Section S5 to include these results.

3. What is the loading of SP? Only “70 wt. % PMMA + SP” was mentioned, and the exact composition is unclear.

We thank reviewer for their comment. We have now modified the manuscript and supporting information to clarify that our blends contain 70 wt.% PMMA + 30 wt.% SP.

4. It was shown that the thermal-back reaction could be performed at 60°C, but is it possible to carry out the UV-induced switching also at high temperature (e.g. Section S5)? This experiment would give a more complete picture of the thermal robustness.

We thank reviewer for this insightful comment on investigating the photochromic behavior at elevated temperature. We have now conducted additional experiments to investigate the photochromic behavior of 70 wt.% PMMA + 30 wt.% SP upon UV irradiation at 60°C. Our results indicate a color change from pale yellow (before UV exposure) to violet (after UV exposure) at 60°C, and no significant difference in the color morphing kinetics at 60°C compared to room temperature.

We have now modified the manuscript and supporting information Section S5 to include these results.

5. Would the chemical shielding resist oxidative conditions (such as a H₂O₂ solution)?

We thank reviewer for this insightful comment on the chemical resistance of our superomniphobic surfaces to oxidative conditions. We have now conducted additional experiments to evaluate the photochromic behavior of our fabrics coated with PMMA + SP + FDTES blends, before and after exposure to 30% H₂O₂ solution. The results indicate that our fabrics retain their color morphing properties even after exposure to harsh oxidative conditions of 30% H₂O₂ solution because of the chemical shielding offered by superomniphobicity.

We have now modified the manuscript and included a new Movie S5 to demonstrate these results.

6. Regarding the fatigue resistance of this system, the authors described that the color morphing remained effective for 10 cycles, but no experimental data was provided in the SI.

We thank reviewer for this insightful comment on retention of color morphing after multiple cycles. We have now included the experimental data (absorption spectra, contact angles and sliding angles) to confirm that our surfaces are indeed effective for 10 cycles without loss in color morphing or superomniphobicity.

We have now modified the supporting information Section S5 to demonstrate these results.

7. In Figure 5 for example, the water and hexadecane have obviously been dyed. The identity and concentration of these dyes should be given in the Methods section.

We thank the reviewer for this comment. We have now modified the methods section and figure captions to clarify that we used 10 µg/mL methylene blue to dye water and 10 µg/mL Oil Red O to dye hexadecane.

8. What are the identities of thermochromic pigments utilized for the studies in Figure 5?

We thank the reviewer for this comment. The photochromic and thermochromic pigments used in this work were procured from Uniglow Pigments and Atlanta Chemical Engineering, respectively. We have now included the model numbers for the pigments in the Methods section.

9. In Figure 1, the chemical structure of FDTES should be drawn in the correct orientation and with better resolution.

We thank the reviewer for this comment. We have now modified Figure 1 of the manuscript to include the chemical structure of FDTES in the horizontal orientation.

We thank the reviewer for carefully reading our manuscript, recognizing the novelty and potentially supporting the publication of our work in *Nature Communications*. We have now substantially revised the manuscript based on the reviewer's suggestions. We sincerely hope that the reviewer will find our revisions satisfactory.

Response to Reviewer #2

This manuscript reported color morphing surfaces with effective chemical shielding. The authors carried out the studies about the effects of polymer polarity, PMMA content and FDTES content, etc. After careful reading of this manuscript, I do not think it can be accepted for publication in the journal mainly because of its low novelty as listed below. Instead, I think Scientific Report is a suitable journal for the study.

We thank the reviewer for carefully reading our manuscript. We have now conducted an extensive set of additional experiments and substantially revised the manuscript and supporting information based on the reviewer's suggestions and better clarified the novelty of our work. We sincerely hope that the reviewer will find our revisions satisfactory and our work suitable for publication in *Nature Communications*.

1. The authors mentioned in the Introduction that "Prior studies have indicated that polymers with polar functional groups display favorable dipole-dipole interactions with the MC form.^{39,40}". So, it is easy to combine polar polymers like PMMA with SP. The authors did not show anything new regarding the photochromic materials.

We thank the reviewer for this comment regarding the novelty of our work. We agree with the reviewer that we are not the first to come up with photochromic materials. However, that is not the novelty of our work.

The primary novelty of our work lies in elucidating the design rationale and careful materials selection to fabricate photochromic materials that retain their photochromism even when exposed to harsh chemical environments – this is not possible with conventional photochromic materials (including references 39 and 40) because they degrade easily when exposed to harsh chemical environments. Accomplishing such multifunctionality requires combining two different surface properties – color morphing (via photochromic or thermochromic materials) and chemical shielding (via superomniphobicity). Such color morphing surfaces with effective chemical shielding (e.g., photochromic and superomniphobic surfaces) have never been reported before (also explained in response to the next comment). By imparting superomniphobicity to photochromic materials, our work enables the utility of photochromic materials even when exposed to harsh chemical environments. Furthermore, by imparting photochromism to chemical shielding materials/fabrics, our work paves the path to practical applications (e.g., camouflage of soldier apparel used in chem-bio warfare), which are not possible with conventional photochromic materials. So, we humbly appeal to the reviewer that our work on combining photochromism or thermochromism with superomniphobicity is novel.

Accomplishing such multifunctionality may appear uncomplicated after we explained the design rationale clearly and demonstrated the properties experimentally. However, we would like to emphasize that our materials selection/design is neither trivial nor obvious – combining color morphing and chemical shielding required exhaustive experiments and rigorous characterization. For instance, it is not just a trivial combination of PMMA + SP at any random composition – the color morphing kinetics required a thorough investigation to identify the appropriate composition. More importantly, the composition of the ternary system – PMMA + SP + FDTES – required a

thorough investigation to achieve multifunctionality. In the ternary system, the choice of using FDTES instead of the more common and more reactive fluorodecyl trichlorosilane as the low surface energy material is not obvious until the materials selection is described clearly, and the properties demonstrated experimentally. Furthermore, it is not trivial that the entire ternary material system (PMMA + SP + FDTES) must be soluble in acetone (common solvent) – there are many polar polymers that do not satisfy this criterion – this is critical for enabling spray-coating, which provides re-entrant texture that in turn is mandatory for superomniphobicity. Based on all these aspects, we humbly appeal to the reviewer that there are many non-trivial and non-obvious aspects in our work, which contribute to making our color morphing surfaces with effective chemical shielding truly novel.

2. The authors mentioned in the Introduction that “While there are prior reports on imparting superhydrophobicity (i.e., extreme repellency to high surface tension liquids like water),²²⁻²⁵ to the best of our knowledge, there are no reports on imparting superomniphobicity (i.e., extreme repellency to both high and low surface tension liquids) to photochromic materials.” After careful reading of the references, I found that Ref. 23 has already reported superomniphobic photochromic materials. The superamphiphobicity was tested using much smaller droplets of 5 μ L compared with this work.

We thank the reviewer for this comment. We are thoroughly familiar with reference 23 and deliberately included it in our manuscript. We would like to reemphasize that there are no reports on imparting superomniphobicity to photochromic materials, as explained further below.

Firstly, reference 23, and many similar reports in literature (not related to photochromism or thermochromism), erroneously claim superomniphobicity based only on high contact angles, regardless of the sliding angle (a measure of slipperiness). As the reviewer may already know, superomniphobic surfaces must not only display very high contact angles, but also display very low sliding angles (typically $< 10^\circ$) for both high and low surface tension liquids (*Tuteja et al., Science, 2007; Pan et al., J. Am. Chem. Soc., 2013; Golovin et al., Angewandte Chemie, 2013; Liu et al., Science, 2014; Kota et al., NPG Asia Mater., 2014; Yong et al., Chem Soc. Rev., 2017; Wang et al., Adv. Mater., 2017; Vahabi et al., Sci. Adv., 2018; Yun et al. Sci. Adv., 2018; Wang et al., Nat. Commun., 2019*). In reference 23, the surfaces displayed a sliding angle of 23° with a 5 μ L droplet of hexadecane (a representative liquid with low surface tension, $\gamma_v \approx 27.5$ mN/m), signifying lack of slipperiness, and consequently, lack of superomniphobicity. In contrast to reference 23, our surfaces not only displayed high apparent contact angles, but also a low sliding angle of 7° with a 20 μ L droplet of hexadecane, indicating high slipperiness, and consequently, superomniphobicity.

Secondly, based on the reviewer’s valuable suggestion, we have now conducted additional experiments to measure the sliding angles with 5 μ L droplets of hexadecane. On smooth glass slides (not textured fabrics) thoroughly spray coated with our PMMA + SP + FDTES blends, a 5 μ L droplet of hexadecane displayed a low sliding angle of 9° , indicating high slipperiness. Furthermore, a 5 μ L droplet of dodecane (with an even lower surface tension, $\gamma_v \approx 25$ mN/m) also displayed a low sliding angle of 10° , indicating high slipperiness. These results clearly indicate that the surfaces reported in our work are superomniphobic, even with 5 μ L droplets, in striking contrast to reference 23.

We have now modified the manuscript to indicate that our surfaces are superomniphobic even with 5 μL droplets, in addition to 20 μL droplets. In summary, we would like to reemphasize that there are no reports on imparting superomniphobicity to photochromic materials.

3. I found that the authors carried most of their studies on rough substrates like fabrics and paper. This is unreasonable to carry out such studies on rough substrates. It is well known in the field that it is easy to form superamphiphobic coatings on rough substrates compared with flat ones like glass slides.

We thank the reviewer for this comment, which perhaps arose because our description of the substrates used in this work was unclear. We would like to clarify that majority of the substrates used in this work are smooth substrates (e.g., glass slides in Figure 3, as well as acrylic, PET, aluminum, copper and tin in Figures 4h-4l); paper and fabrics are the only rough substrates. Here, we would like to emphasize that spray coating the optimized PMMA + SP + FDTES blends on smooth substrates already renders them superomniphobic. The enhancement in superomniphobicity due to roughness of the substrate (e.g., paper or fabric) is minimal. In summary, spray coating our optimized PMMA + SP + FDTES blends renders substrates photochromic and superomniphobic, regardless of whether they are smooth or rough.

We have now modified the manuscript to better clarify which substrates are smooth and rough.

4. During measurements of superamphiphobicity, the authors used very large droplet volume of 20 μL . It is well known again in the field that larger droplet volume is helpful to record lower sliding angle. Such large volume is seldom used in the field.

We thank the reviewer for this important comment, which has helped us improve our manuscript. Based on the reviewer's valuable suggestion, we have now conducted additional experiments to measure the sliding angles with 5 μL droplets. On smooth glass slides (not textured fabrics) thoroughly spray coated with our PMMA + SP + FDTES blends, a 5 μL droplet of hexadecane displayed a low sliding angle of 9° , indicating high slipperiness. Furthermore, a 5 μL droplet of dodecane (with an even lower surface tension, $\gamma_v \approx 25 \text{ mN/m}$) also displayed a low sliding angle of 10° , indicating high slipperiness. These results clearly indicate that the surfaces reported in our work are superomniphobic, even with 5 μL droplets.

We have now modified the manuscript to indicate that our surfaces are superomniphobic even with 5 μL droplets, in addition to 20 μL droplets.

5. It is also well known that spray-coating can easily form superamphiphobic coatings on various substrates (e.g., Applied Surface Science 2017, 400, 162).

We thank the reviewer for this comment. We agree with the reviewer that spray coating allows fabrication of superomniphobic surfaces. In fact, our research group has significant expertise in spray coating and has used it extensively to make a wide range of superhydrophobic and superomniphobic surfaces (Wang, et al., Mater. Horiz., 2022; Hatoum et al, J Mech Behav Biomed Mater., 2020; Movafaghi et al., Adv. Mater., Interfaces 2019; Vahabi et al., Sci. Adv., 2018;

Pendurthi et al., ACS Appl Mater Interfaces, 2017; Vahabi et al., ACS Appl Mater Interfaces, 2016; Vallabhuneni et al., Macromol Mater Eng., 2018; Wang et al., ACS Appl Mater Interfaces, 2016). Here, we would like to clarify that we chose spray coating in this work because it is a simple, scalable and economical technique that provides the re-entrant texture required for superomniphobicity. However, that is not the primary novelty of our work. The primary novelty of our work lies in elucidating the design rationale and careful materials selection to fabricate color morphing surfaces that retain their color morphing properties even when exposed to harsh chemical environments, by imparting superomniphobicity. Such color morphing surfaces with effective chemical shielding have never been reported before.

6. Moreover, there is no in-depth discussion and new finding regarding superamphiphobic materials and photochromic materials throughout the manuscript.

We thank the reviewer for this comment. We agree with the reviewer that we are not reporting a new finding independently in just superomniphobicity or in just photochromism. Both of these properties are independently well understood, and our research groups have extensive prior experience working in each of these areas. However, a combination of these two properties – photochromism and superomniphobicity – is neither reported before nor trivial/obvious. The primary novelty of our work lies in elucidating the design rationale and careful materials selection to fabricate color morphing surfaces that retain their color morphing properties even when exposed to harsh chemical environments, by imparting superomniphobicity.

There are also other concerns as shown below, which may be helpful for the authors to improve the manuscript before submitting it elsewhere.

1. The authors mentioned in the abstract that “but almost all photochromic materials suffer from degradation when exposed to moist/humid environments or harsh chemical environments”. However, the authors did not perform any study about the resistance of their photochromic materials when exposed to moist/humid environments.

We thank the reviewer for this comment. Based on the reviewer’s suggestion, we have now conducted additional experiments by exposing our photochromic and superomniphobic surfaces to air with ~50% relative humidity and immersing them under water. Our results indicate that the surfaces retained both photochromism and superomniphobicity after 10 days of continuous air exposure and water immersion. As the reviewer may already know, resistance to moist or humid conditions is much easier for any superomniphobic surface compared to significantly harsher conditions like immersion in 6M sulfuric acid (Movie S4) or 30% hydrogen peroxide (new Movie S5).

We have now modified the manuscript and supporting information Section S12 to demonstrate the results.

2. It is clear that the fabrics were coated with a lot of the coating materials (Figure 4b). This will inevitably affect the breathability.

We thank the reviewer for this comment on breathability. Prior to submitting this manuscript, we had already used a capillary flow porometer to measure the air flow rate through uncoated and coated fabrics at different pressure differentials. Our results indicated negligible differences in flow rate vs. pressure differential data for uncoated and coated fabrics, indicating the retention of breathability even after coating. As evident from Figure 4b, although the surfaces of individual fibers are covered by the coating, the large pores (on the order of 100 μm) between the fiber bundles remain wide open, allowing air to permeate easily (through the path of least resistance) without influencing the flow rate significantly.

We have now modified the manuscript and included a new supporting information Section S11 to demonstrate these results.

3. How about stability of the photochromic materials? Is there any changes in the color parameters and photochromic dynamics after for example hundreds of coloring-decoloring cycles?

We thank the reviewer for this comment about the number of color morphing cycles.

Upon exposure to harsh chemical conditions, our photochromic superomniphobic materials easily outperform photochromic materials without superomniphobicity in terms of color morphing cycles. For example, the Spiropyran-based photochromic materials without superomniphobicity cannot display even a single color morphing cycle after exposure to harsh chemical conditions (e.g., concentrated sulfuric acid or hydrogen peroxide). In contrast, our Spiropyran-based photochromic superomniphobic materials can easily display 10 color morphing cycles after exposure to the same harsh chemical conditions (see new data in revised SI Section S5). As the reviewer may already know, under ambient conditions (i.e., even without harsh chemicals), Spiropyran to Merocyanine transformation is known to display photochromic fatigue (i.e., loss in color) after about 10 color morphing cycles (*Kundu et al., Nature Chemistry, 2015; Putri et al., Chem Phys Chem., 2016; Klajn et al., Chem. Soc. Rev., 2014; He et al., Colloids and Surfaces A, 2020; Yang et al., Langmuir, 2018; Maculi et al., Journal of Physics: Conference Series, 2009; Radu et al., J. Photochem. Photobiol., A, 2009*). Imparting superomniphobicity does not deteriorate the number of color morphing cycles under ambient conditions, but it greatly improves the number color morphing cycles when exposed to harsh chemicals.

As another example, we have now conducted additional experiments with commercial photochromic pigments, which can undergo hundreds of color morphing cycles, without noticeable photochromic fatigue under ambient conditions. When such photochromic pigments without superomniphobicity are exposed to harsh chemical conditions (e.g., concentrated sulfuric acid or hydrogen peroxide), they cannot display even a single color morphing cycle. In striking contrast, such color morphing pigments with superomniphobicity can display hundreds of color morphing cycles after exposure to the same harsh chemical conditions. This clearly exemplifies the chemical shielding (or chemical stability) that addition of superomniphobicity imparts to existing photochromic materials so that they can retain photochromism (e.g., color morphing cycles) even when exposed to harsh chemical conditions.

We have now modified the manuscript and included a new supporting information Section S14 to demonstrate these results.

4. How about stability of the superamphiphobicity? As a superamphiphobic coating on fabric, Martindale abrasion and washing stability tests are essential.

We thank the reviewer for this comment about the durability of superomniphobic surfaces. As the reviewer may already know, improving the mechanical durability (e.g., abrasion against solids etc.) of super-repellent surfaces continues to be a grand challenge in the field of surface science (Chu *et al.*, *Chem. Soc. Rev.*, 2014; Kota *et al.*, *NPG Asia Mater.*, 2014; Tian *et al.*, *Science*, 2016; Yong *et al.*, *Chem. Soc. Rev.*, 2017; Chen *et al.*, *J Mater. Chem. A*, 2017; Ai *et al.*, *Chem. Comm.*, 2019). Indeed, efforts to improve the mechanical durability of superomniphobic surfaces are underway in our research group as well as other research groups across the world. However, improving mechanical durability of superomniphobic surfaces (while important for product development) is not the primary focus of this work. The primary focus of our work is to elucidate the design rationale and careful materials selection for fabrication of novel color morphing materials that can retain their color morphing properties even when exposed to harsh chemical environments.

Having said that, based on the reviewer's suggestion, we have now conducted additional experiments to assess the durability of our photochromic superomniphobic surfaces. Our photochromic superomniphobic surfaces retain both photochromism and superomniphobicity after immersion in water for 10 days, after sliding thousands of water and hexadecane droplets, and against light abrasion due to touch, but they do not have sufficient durability to withstand severe abrasion against solids (e.g., Taber or Martindale abrasion with sandpapers). We have now revised the manuscript and supporting information to include these results and clearly indicate that our photochromic superomniphobic surfaces are not durable against solid abrasion and future work is necessary to address this grand challenge.

Overall, we thank the reviewer for carefully reading our manuscript. The reviewer's comments have greatly helped us improve the manuscript. We have now conducted an extensive set of additional experiments and substantially revised the manuscript and supporting information based on the reviewer's suggestions and better clarified the novelty of our work. We sincerely hope that the reviewer will find our revisions satisfactory and our work suitable for publication in *Nature Communications*.

Response to Reviewer #3

This is a well written manuscript on a really interesting topic. Photochromic and thermochromic dyes have been utilized for a wide variety of applications for many years. More recently, the field has been particularly interested in the design and utilization of polar dyes, such as spiropyran used here, as such dyes can sustain their photochromic properties, and may also display a faster response time. The challenge with such dyes is that they can readily lose their performance when they are utilized in an environment where other ions / polar molecules are present. This is the exact challenge that this work aims to address via the utilization of omniphobic surfaces, and this is a major achievement in this field in my opinion. The work performed is easy to understand, and is well supported by fundamental performance / data analysis.

We thank the reviewer for carefully reading our manuscript, recognizing the novelty and achievement of our work. We have now revised the manuscript based on the reviewer's suggestions. We sincerely hope that the reviewer will find our revisions satisfactory.

I have a few minor issues that should be addressed:

1. It would be good to report the Hansen polar solubility parameter for both SP and MC in order to compare with PDMS and PMMA.

We thank the reviewer for this comment. We have now modified the manuscript to include the Hansen polar solubility parameters for SP ($\delta_p \sim 7.8$; Abdollahi et al., *J. Mater. Chem. C*, 2017) and MC ($\delta_p \sim 7.6$; Abdollahi et al., *J. Mater. Chem. C*, 2017). The proximity of these Hansen polar solubility parameters to that of PMMA ($\delta_p \sim 10.5$; Hansen, *Hansen solubility parameters: A user's handbook*, CRC press, 2007) than that of PDMS ($\delta_p \sim 2.5$; Hansen, *Hansen solubility parameters: A user's handbook*, CRC press, 2007) possibly indicates that SP and MC are more compatible with PMMA than PDMS.

2. The commercial specific photochromic / thermochromic dyes utilized for the experiments shown in Fig. 5 were not clear to me. Those should be clarified in the text and added to the materials and methods section.

We thank the reviewer for this comment. The photochromic and thermochromic pigments used in this work were procured from Uniglow Pigments and Atlanta Chemical Engineering, respectively. We have now included the model numbers for the pigments in the Methods section.

Once again, we thank the reviewer for carefully reading our manuscript, recognizing the novelty and potentially supporting the publication of our work in *Nature Communications*. We have now revised the manuscript based on the reviewer suggestions. We sincerely hope that the reviewer will find our revisions satisfactory.

REVIEWER COMMENTS

Reviewer #1 (Remarks to the Author):

The authors have addressed my questions and concerns, and I would be happy to recommend this manuscript for publication.

Reviewer #2 (Remarks to the Author):

I am very happy to discuss novelty of this study with the authors again.

- Regarding response to my Question 2:

As is well known, there are two kinds of superhydrophobic surfaces, the lotus-inspired ones (with high contact angle and low sliding angle) and rose petal-inspired ones (with high contact angle and stably adhesion of droplets on the surfaces). Superamphiphobic surfaces are also the same. The sliding angle is not necessarily to be very low. Compared with Ref. 23, the authors really further reduced the sliding angle to $<10^\circ$. I would like to say this is an improvement but not a good innovation. On the basis of Ref. 23, it is easy to get the idea of further enhancing the superamphiphobicity.

- Regarding response to my Question 5:

The authors mentioned "The primary novelty of our work lies in elucidating the design rationale and careful materials selection to fabricate color morphing surfaces that retain their color morphing properties even when exposed to harsh chemical environments, by imparting superomniphobicity. Such color morphing surfaces with effective chemical shielding have never been reported before."

So, I further explored literature on this topic and found a paper from the same group of Ref. 23. In the paper (ACS Appl. Mater. Interfaces 2017, 9, 1941–1952), the authors reported combination of pigments and superamphiphobicity. As wrote in the abstract, "The colorful superamphiphobic coatings feature high contact angles and low SAs for various liquids, including water and n-decane. The coatings also showed high mechanical, environmental, chemical, and thermal durability even under harsh conditions." Although the pigments are not photochromic, the authors have already give the idea of using superamphiphobicity to enhance the stability of pigments. Following this paper, two years later the authors reported Ref. 23 (using superamphiphobicity to enhance the stability of photochromic materials). In fact, this group first reported colorful superhydrophobic coatings, and then to colorful superamphiphobic ones, and then to photochromic superamphiphobic ones step by step.

So, I still think the novelty of this work here is not sufficient for publication in Nature Communications.

Response to Reviewer#1

The authors have addressed my questions and concerns, and I would be happy to recommend this manuscript for publication.

We thank the reviewer for carefully reading our manuscript, recognizing the novelty and supporting the publication of our work in *Nature Communications*.

Response to Reviewer #2

I am very happy to discuss novelty of this study with the authors again.

We thank the reviewer for carefully reading our manuscript again. The reviewer's thoughtful comments in the previous round of reviews have helped us improve the manuscript significantly, and we are grateful to the reviewer for that. We also respect the creativity in publications from Prof. Junping Zhang's group (like Ref 23, *Dong et al., ACS Appl. Mater. Interf.* 2017 etc.).

We have now attempted to better clarify our thought process and the novelty of our work. We sincerely hope that the reviewer will find our novelty satisfactory and our work suitable for publication in *Nature Communications*.

- Regarding response to my Question 2:

As is well known, there are two kinds of superhydrophobic surfaces, the lotus-inspired ones (with high contact angle and low sliding angle) and rose petal-inspired ones (with high contact angle and stably adhesion of droplets on the surfaces). Superamphiphobic surfaces are also the same. The sliding angle is not necessarily to be very low.

As the reviewer may already know, contact angle is a measure of the liquid repellency; contact angle hysteresis or sliding angle is a measure of the slipperiness. A surface is considered super-repellent (e.g., superhydrophobic or superomniphobic) when it displays very high liquid repellency (typically contact angles $> 150^\circ$) **and** very high slipperiness (typically contact angle hystereses $< 10^\circ$ or sliding angles $< 10^\circ$). Achieving such super-repellent surfaces requires the liquid droplets to adopt the Cassie-Baxter state (*Cassie et al., Trans. Faraday Soc.* 1944) on a solid surface with a high fraction of liquid-air interfacial area.

Unfortunately, there has been a misconception that surfaces (like rose petals) with very high liquid repellency (i.e., contact angles $> 150^\circ$) can be considered super-repellent even though they lack slipperiness (i.e., sliding angles $> 10^\circ$). Such surfaces (with very high liquid repellency, but lacking slipperiness) can be achieved when the liquid droplets adopt the Wenzel state (*Wenzel et al., Ind. Eng. Chem. Res.* 1936) partially or completely – they are different from super-repellent surfaces (with very high liquid repellency and very high slipperiness), which require the Cassie-Baxter state with a high fraction of liquid-air interfacial area.

There are innumerable reports from many prestigious research groups across the world reinforcing that super-repellent surfaces display very high liquid repellency (typically contact angles $> 150^\circ$) **and** very high slipperiness (typically contact angle hystereses $< 10^\circ$ or sliding angles $< 10^\circ$). Here are a few examples to illustrate/imply this:

Dr. Hans-Jürgen Butt & Dr. Doris Vollmer's group from Max Planck Institute, Germany

On superamphiphobic layers, even nonpolar liquids form an apparent contact angle above 150° and a roll-off angle below 10° in air.

(Small 2023; Nano Lett. 2023; Adv.Mater. 2019; Adv. Mater. 2018; Science 2012)

Dr. Stefan Seeger's group from the University of Zurich, Switzerland

Superhydrophobic and superoleophobic surfaces have contact angles greater than 150° and low contact angle hysteresis not only towards probing water but also for low surface tension oils. (Small 2022; Chem. Soc. Rev. 2014; Angew. Chem. Int. Ed. 2011; Adv. Mater. 2006)

Dr. Pavel Levkin's group from Karlsruhe Institute of Technology, Germany

Surfaces that display extreme liquid repellency, characterized by their large apparent liquid contact angle ($\theta^ > 150^\circ$) and a small roll-off angle (α) or contact angle hysteresis ($\Delta\theta^*$), are especially interesting from the practical perspective but at the same time challenging to fabricate. (Adv. Funct. Mater. 2023; Adv. Mater. 2021; Adv. Mater. 2013; Adv. Mater. 2018)*

Dr. Robin Ras's group from Aalto University, Finland

Water droplets that contact these surfaces must have large apparent contact angles (greater than 150 degrees) and small roll-off angles (less than 10 degrees). (Adv. Mater. 2023; Nature 2020; Science 2016; Adv. Mater. 2011)

Dr. Ivan Parkin's group from University College London, UK

Superhydrophobic surfaces are those that exhibit water contact angles greater than 150° and sliding angle less than 10°. (Chem. Soc. Rev., 2022; ACS Nano 2018; Science 2015; J. Mater. Chem. A, 2013)

Dr. Anish Tuteja's group from the University of Michigan, USA

Superomniphobic surfaces display contact angles $>150^\circ$ and low contact angle hysteresis with essentially all contacting liquids. (J. Am. Chem. Soc. 2013; Science 2007; Angew. Chem. Int. Ed. 2013)

Dr. Xu Deng's group the University of Electronic Science and Technology, China

Superhydrophobic surfaces display apparent contact angles with water greater than 150° and low contact angle hysteresis. (Adv. Mater. 2022; Chem. Soc. Rev. 2021; Adv. Sci. 2020)

Dr. Atsushi Hozumi's group from Institute of Advanced Industrial Science & Technology, Japan

Superhydrophobic materials with extremely high contact angles (CAs, more than 150°), allowing a small volume water droplet to move at low sliding angles (less than 5–10°), have been extensively studied in the past few decades. (Langmuir 2021; Langmuir 2018; J. Mater. Chem. A, 2017; ACS Appl. Mater. Interf. 2015)

Based on all of the above, it is evident that low sliding angles (an indication of high slipperiness) are required to qualify a surface as super-repellent. Achieving a superomniphobic surface with a low sliding angle is non-trivial because it requires the Cassie-Baxter state with a high fraction of liquid-air interfacial area, especially with low surface tension liquids. And achieving the Cassie-Baxter state with low surface tension liquids is also non-trivial because it requires an appropriate combination of low surface energy and re-entrant texture, as described in our manuscript. Furthermore, accomplishing multifunctionality (photochromism or thermochromism in addition

to superomniphobicity) makes the materials design, selection and processing even more restrictive, and non-trivial.

Compared with Ref. 23, the authors really further reduced the sliding angle to $<10^\circ$. I would like to say this is an improvement but not a good innovation.

First, we would like to submit that we are familiar with and respect the creativity in Ref 23 and many other publications from Prof. Junping Zhang's group. Next, we would like to humbly submit that our work has many novel aspects compared to Ref 23 (including the non-trivial reduction in sliding angles). For example, unlike Ref 23:

- We demonstrated color morphing surfaces with effective chemical shielding against exposure to corrosive liquids (8M H₂SO₄), oxidizers (30% H₂O₂) and humid air.
- We demonstrated reversible photochromism for hundreds of cycles without loss in color morphing or superomniphobicity.
- We demonstrated thermochromism along with superomniphobicity.
- We demonstrated color morphing surfaces with effective chemical shielding on many smooth substrates (e.g., glass, polymers, metals) and rough substrates (paper and fabrics).
- We demonstrated the role of polymer polarity in color morphing kinetics.
- We demonstrated a non-trivial reduction in sliding angles of hexadecane (from 26° to 9° for 5 μL droplets).

We sincerely hope that the reviewer will find our novelty and unique contributions satisfactory and our article suitable for publication in *Nature Communications*.

On the basis of Ref. 23, it is easy to get the idea of further enhancing the superamphiphobicity.

We are not denying inspiration and learning from prior work – as diligent students of science, all of us seek to learn from prior work every day – that is why we cited relevant prior work (47 references, including Ref 23) in our manuscript. However, we would like to clarify that getting inspired by prior work is very different from accomplishing the idea, demonstrating the properties experimentally and explaining the design rationale clearly.

In our work, accomplishing multifunctionality by simultaneously combining two surface properties – color morphing (via photochromic or thermochromic materials) and chemical shielding (via superomniphobicity) – may appear uncomplicated after we explained the design rationale clearly and demonstrated the properties experimentally. However, we would like to emphasize that our materials selection/design is neither trivial nor obvious – combining color morphing and chemical shielding required exhaustive experiments and rigorous characterization. For instance, it is not just a trivial combination of PMMA + SP at any random composition – the color morphing kinetics required a thorough investigation to identify the appropriate composition. More importantly, the composition of the ternary system – PMMA + SP + FDTES – required a thorough investigation to achieve multifunctionality. In the ternary system, the choice of using FDTES instead of the more common and more reactive fluorodecyl trichlorosilane as the low surface energy material is not obvious until the materials selection is described clearly, and the properties demonstrated experimentally. Furthermore, it is not trivial that the entire ternary

material system (PMMA + SP + FDTES) must be soluble in acetone (common solvent) – there are many polar polymers that do not satisfy this criterion – this is critical for enabling spray-coating, which provides re-entrant texture that in turn is mandatory for superomniphobicity.

Based on all these (and the list of novel aspects compared to Ref 23 presented in response to the previous comment above), we humbly appeal to the reviewer that there are many non-trivial, non-obvious and unique aspects in our work, which contribute to making our color morphing surfaces with effective chemical shielding truly novel. We sincerely hope that the reviewer will find our novelty and unique contributions satisfactory and our article suitable for publication in *Nature Communications*.

- Regarding response to my Question 5:

The authors mentioned “The primary novelty of our work lies in elucidating the design rationale and careful materials selection to fabricate color morphing surfaces that retain their color morphing properties even when exposed to harsh chemical environments, by imparting superomniphobicity. Such color morphing surfaces with effective chemical shielding have never been reported before.”

So, I further explored literature on this topic and found a paper from the same group of Ref. 23. In the paper (ACS Appl. Mater. Interfaces 2017, 9, 1941–1952), the authors reported combination of pigments and superamphiphobicity. As wrote in the abstract, “The colorful superamphiphobic coatings feature high contact angles and low SAs for various liquids, including water and n-decane. The coatings also showed high mechanical, environmental, chemical, and thermal durability even under harsh conditions.” Although the pigments are not photochromic, the authors have already give the idea of using superamphiphobicity to enhance the stability of pigments. Following this paper, two years later the authors reported Ref. 23 (using superamphiphobicity to enhance the stability of photochromic materials). In fact, this group first reported colorful superhydrophobic coatings, and then to colorful superamphiphobic ones, and then to photochromic superamphiphobic ones step by step.

We would like to reiterate that we are familiar with and respect the creativity in publications from Prof. Junping Zhang’s group (like Ref 23, *Dong et al., ACS Appl. Mater. Interf., 2017* etc.). Unlike Ref 23, the work by *Dong et al., ACS Appl. Mater. Interf., 2017* (newly cited by the reviewer) is not directly relevant to our work because it presents passive colored surfaces without any active color morphing (photochromism or thermochromism). The reviewer also acknowledged that “*pigments are not photochromic*” in the work by *Dong et al., ACS Appl. Mater. Interf., 2017*. And we would like to reemphasize that the work by *Dong et al., ACS Appl. Mater. Interf., 2017* was neither an inspiration for nor directly related to our work.

Having said that, we are not denying inspiration and learning from other prior work – as diligent students of science, all of us seek to learn from prior work every day – that is why we cited relevant prior work (47 references, including Ref 23) in our manuscript. However, we would like to clarify that getting inspired by prior work is very different from accomplishing the idea, demonstrating the properties experimentally and explaining the design rationale clearly.

While Ref 23 is one of the many works that have inspired us, we would like to humbly submit once again that our work has many novel aspects compared to Ref 23 (including the non-trivial reduction in sliding angles). For example, unlike Ref 23:

- We demonstrated color morphing surfaces with effective chemical shielding against exposure to corrosive liquids (8M H₂SO₄), oxidizers (30% H₂O₂) and humid air.
- We demonstrated reversible photochromism for hundreds of cycles without loss in color morphing or superomniphobicity.
- We demonstrated thermochromism along with superomniphobicity.
- We demonstrated color morphing surfaces with effective chemical shielding on many smooth substrates (e.g., glass, polymers, metals) and rough substrates (paper and fabrics).
- We demonstrated the role of polymer polarity in color morphing kinetics.
- We demonstrated a non-trivial reduction in sliding angles of hexadecane (from 26° to 9° for 5 μL droplets).

We sincerely hope that the reviewer will find our novelty and unique contributions satisfactory and our article suitable for publication in *Nature Communications*.

So, I still think the novelty of this work here is not sufficient for publication in Nature Communications.

We would like to reiterate that the primary novelty of our work lies in elucidating the design rationale and careful materials selection to fabricate color morphing surfaces that retain their color morphing properties even when exposed to harsh chemical environments, by imparting superomniphobicity. Accomplishing such multifunctionality requires combining two different surface properties – color morphing (via photochromic or thermochromic materials) and chemical shielding (via superomniphobicity). While such multifunctionality may appear uncomplicated after we explained the design rationale clearly and demonstrated the properties experimentally, we would like to emphasize that literature is rife with articles on multifunctional surfaces (combining super-repellency with another property) in high impact journals because of the importance of such multifunctional surfaces. Here are a few examples:

Super-repellent and flame retardant

Qu et al., Adv. Funct. Mater., 2023; Wang et al., Adv. Mater., 2022; Song et al., ACS Nano, 2021; Sun et al., ACS Nano, 2015; Lu et al., Angew. Chem., 2014.

Super-repellent and photothermal

Zheng et al., Adv. Mater., 2024; Jiang et al., Adv. Sci., 2023; Chen et al., Adv. Funct. Mater., 2023; Lai et al., Adv. Mater., 2023; Sun et al., Adv. Mater., 2022, 34, 2108232; Mitridis et al., ACS Nano, 2020; Hu et al., ACS Nano, 2015

Super-repellent and electrically conductive

Lou et al., Angew. Chem., 2023; Das et al., Mater. Horiz., 2021; Wang et al., Sci. Adv., 2017; Zhai et al., Adv. Mater., 2008; Lee et al., Adv. Mater., 2008; Zhang et al., Adv. Funct. Mater., 2006

Super-repellent and flexible/stretchable

Kong et al., ACS Nano, 2024; Zhou et al., Adv. Mater., 2022; Dai et al., Adv. Funct. Mater., 2021; Wang et al., Adv. Mater., 2020; Sun et al., Adv. Funct. Mater., 2018; Liu et al., Adv. Mater., 2020; Xu et al., Adv. Sci., 2018; Wang et al., Adv. Mater., 2016; Mates et al., Nat Commun., 2015; Cho et al., Adv. Funct. Mater., 2013

Super-repellent and transparent

Li et al., Adv. Funct. Mater., 2024; Chen et al., Adv. Sci., 2021; Wu et al., ACS Nano, 2018; Tenjimbayashi et al., Adv. Funct. Mater., 2016; Yu et al., ACS Nano, 2016; Deng et al., Science, 2012

Super-repellent and shape memory

Chen et al., Adv. Funct. Mater., 2023; Li et al., Adv. Funct. Mater. 2021; Cheng et al., Adv. Mater., 2021; Liu et al., Adv. Sci., 2020; Cheng et al., Adv. Funct. Mater., 2018; Wang et al., Adv. Mater., 2017; Wang et al., Sci. Adv., 2017; Yao et al., Angew. Chem., 2014

Based on all of the above, we humbly appeal to the reviewer that our multifunctional surfaces that combine color morphing with effective chemical shielding are truly novel, and suitable for publication in *Nature Communications*.

Overall, we thank the reviewer for carefully reading our manuscript again. The reviewer's thoughtful comments in the previous round of reviews have helped us improve the manuscript significantly, and we are grateful to the reviewer for that. We sincerely hope that the reviewer will find our novelty satisfactory and our work suitable for publication in *Nature Communications*.

REVIEWERS' COMMENTS

Reviewer #2 (Remarks to the Author):

-In the response to my concerns, the authors mentioned “Unfortunately, there has been a misconception that surfaces...they are different from super-repellent surfaces”. Please at least clarify the so-called “misconception” and define “super-repellent surfaces” in the manuscript. In fact, in the response to my concerns, the authors used many times “super-repellent surfaces”, but ““super-repellent surfaces”” was not mentioned at all in the manuscript.

-There are many papers about colorful (Ref. 23, ACS Appl. Mater. Interfaces 2017, 9, 1941; SURFACE AND INTERFACE ANALYSIS 2021, 53, 365-373; Appl Surface Sci, 2019, 466 , 328-341;) or photochromic (ADVANCES IN POLYMER TECHNOLOGY 2019, 9536320; CHEMICAL ENGINEERING JOURNAL 2022, 444, 136604) superhydrophobic/superamphiphobic coatings. Their aim, the same as this study, is to enhance the stability of the pigments or photochromic materials by using superhydrophobic/superamphiphobic coatings. So, I do think the novelty of this work is not sufficient for publication in the journal.

The editor may choose to publish it or not, yet I maintain my own perspective.

Response to Reviewer #2

-In the response to my concerns, the authors mentioned “Unfortunately, there has been a misconception that surfaces...they are different from super-repellent surfaces”. Please at least clarify the so-called “misconception” and define “super-repellent surfaces” in the manuscript. In fact, in the response to my concerns, the authors used many times “super-repellent surfaces”, but ““super-repellent surfaces”” was not mentioned at all in the manuscript.

We thank the reviewer for carefully reading our previous set of responses. Based on the reviewer’s comment above, we have now modified the manuscript to clarify what super-repellent surfaces are and what is important in qualifying the super-repellency of a surface.

-There are many papers about colorful (Ref. 23, ACS Appl. Mater. Interfaces 2017, 9, 1941; SURFACE AND INTERFACE ANALYSIS 2021, 53, 365-373; Appl Surface Sci, 2019, 466, 328-341;) or photochromic (ADVANCES IN POLYMER TECHNOLOGY 2019, 9536320; CHEMICAL ENGINEERING JOURNAL 2022, 444, 136604) superhydrophobic/superamphiphobic coatings.

First, we would like to humbly submit that we are familiar with the literature on colorful super-repellent surfaces, such as the following articles cited by the reviewer – Dong *et al.*, ACS Appl. Mater. Interfaces 2017; Qu *et al.*, Surf Interface Anal. 2021. Such articles are not directly relevant to our work because they present passive colored surfaces without any active color morphing (photochromism or thermochromism). And we would like to reemphasize such articles were neither an inspiration for nor directly related to our work.

Second, we would like to humbly submit that we are also familiar with the literature on surfaces that combine superhydrophobicity and photochromism, such as the following articles cited by the reviewer – Dong *et al.*, Journal of Materials Chemistry A 2019; Applied Surface Science 2019; Advances in Polymer Technology 2019; Wan *et al.*, Chemical Engineering Journal 2019. As we mentioned in the manuscript, while there are prior reports on imparting superhydrophobicity, to the best of our knowledge, there are no reports on imparting superomniphobicity to photochromic materials. In the manuscript, we have also clarified what superhydrophobic and superomniphobic surfaces are, and what is important in qualifying the super-repellency of a surface.

Their aim, the same as this study, is to enhance the stability of the pigments or photochromic materials by using superhydrophobic/superamphiphobic coatings. So, I do think the novelty of this work is not sufficient for publication in the journal. The editor may choose to publish it or not, yet I maintain my own perspective.

We would like to reiterate that while there are prior reports on imparting superhydrophobicity, to the best of our knowledge, there are no reports on imparting superomniphobicity to photochromic materials. The primary novelty of our work lies in elucidating the design rationale and careful materials selection to fabricate color morphing surfaces that retain their color morphing properties even when exposed to harsh chemical environments, by imparting superomniphobicity. Accomplishing such multifunctionality requires combining two different surface properties – color

morphing (via photochromic or thermochromic materials) and chemical shielding (via superomniphobicity). While such multifunctionality may appear uncomplicated after we explained the design rationale clearly and demonstrated the properties experimentally, we would like to emphasize that our materials selection/design is neither trivial nor obvious – combining color morphing and chemical shielding required exhaustive experiments and rigorous characterization. For instance, it is not just a trivial combination of PMMA + SP at any random composition – the color morphing kinetics required a thorough investigation to identify the appropriate composition. More importantly, the composition of the ternary system – PMMA + SP + FDTES – required a thorough investigation to achieve multifunctionality. In the ternary system, the choice of using FDTES instead of the more common and more reactive fluorodecyl trichlorosilane as the low surface energy material is not obvious until the materials selection is described clearly, and the properties demonstrated experimentally. Furthermore, it is not trivial that the entire ternary material system (PMMA + SP + FDTES) must be soluble in acetone (common solvent) – there are many polar polymers that do not satisfy this criterion – this is critical for enabling spray-coating, which provides re-entrant texture that in turn is mandatory for superomniphobicity.

We would also like to submit that our work has many novel aspects compared to the references cited by the reviewer that combine superhydrophobicity and photochromism:

- We demonstrated color morphing surfaces with effective chemical shielding against exposure to corrosive liquids (8M H₂SO₄), oxidizers (30% H₂O₂) and humid air.
- We demonstrated reversible photochromism for hundreds of cycles without loss in color morphing or superomniphobicity.
- We demonstrated thermochromism along with superomniphobicity.
- We demonstrated color morphing surfaces with effective chemical shielding on many smooth substrates (e.g., glass, polymers, metals) and rough substrates (paper and fabrics).
- We demonstrated the role of polymer polarity in color morphing kinetics.
- We demonstrated a non-trivial reduction in sliding angles of hexadecane (a representative low surface tension liquid).

Based on all these, we humbly appeal to the reviewer that there are many non-trivial, non-obvious and unique aspects in our work, which contribute to making our color morphing surfaces with effective chemical shielding truly novel. We thank the reviewer for carefully reading our manuscript again, and sincerely hope that the reviewer will find our novelty and unique contributions satisfactory and our article suitable for publication in *Nature Communications*.